# Real-time Anomaly Detection in Epidemic Data Streams

Pulak Agarwal
pulak.agarwal@gatech.edu
Georgia Institute of Technology
Atlanta, Georgia, USA

Pranav Aluru
panualuru@gmail.com
Lakeside High School
Atlanta, Georgia, USA

B. Aditya Prakash
badityap@cc.gatech.edu
Georgia Institute of Technology
Atlanta, Georgia, USA

## ABSTRACT

A key challenge with data collection for time-series data for forecasting is detecting anomalies in real-time. Anomalies can negatively affect the performance of forecasting models, if not accounted for. It is a relatively simple process to retrospectively identify an anomaly in a time-series and correct it, however the problem lies with identifying a particular time-step as anomalous when there is no further information to distinguish it from a change in trend. In the context of the COVID-19 pandemic, accurate forecasts are critical to policymakers, and yet, most data is recorded manually, susceptible to human errors or other technical constraints. While most errors are corrected, it often takes several weeks for these corrections to be made, leading to real-time forecasting models generating predictions on data that is not necessarily accurate. In order to prevent this, we utilize a neural framework called Back2Future and build a simple binary classifier on top of it to propose a real-time anomaly detection system in the context of detecting anomalies in COVID-19 reporting data.

## CCS CONCEPTS

• **Computing methodologies** → **Machine Learning**; **Anomaly Detection**; • **Data Driven Epidemiology**;

## KEYWORDS

Machine learning, time-series data, real-time anomaly detection

**ACM Reference Format:**
Pulak Agarwal, Pranav Aluru, and B. Aditya Prakash. 2022. Real-time Anomaly Detection in Epidemic Data Streams. In *epiDAMIK 2022: 5th epiDAMIK ACM SIGKDD International Workshop on Epidemiology meets Data Mining and Knowledge Discovery, August 15, 2022, Washington, DC, USA.* ACM, New York, NY, USA, 4 pages.

## 1 INTRODUCTION

COVID-19 has caused the deaths of millions of people globally over the span of over 2 years. The wide spread of the disease made it imperative for agencies to report the number of cases and fatalities as a result of COVID-19 in order to ensure proper precautions and interventions were in place. Multiple groups and organizations have been working on collecting data reported by hospitals through government agencies. However, as is often the case with real world time series data, there are often data quality issues. Specifically in the case of COVID-19 case counts, this can be attributed to several

types of issues, including reporting delays, human error, changes in reporting standards, under-reporting and revisions made to data after long periods of time. These issues have been shown to impact the predictions made by statistical forecasting models negatively. [2] It is thus important to identify certain records within the data as anomalous to prevent models from being influenced by incorrect data. There are many studies working on anomaly detection with time series in general, such as Liu et al. [6], and with COVID-19 as well, such as works by Homayouni et al. [3], Francisquini et al. [2], Yom-Tov et al. [8]. However, detecting anomalies in real-time is a challenging task since there is no access to data following the current timestep, and no way of knowing if there will be a rectification in the future that would help identify the original report was anomalous.

We look at the COVID-19 Data Repository by the Center for Systems Science and Engineering (CSSE) at Johns Hopkins University, which is widely regarded as a trusted source for incidence and mortality counts for COVID-19 [1] and use Back2Future [5], a neural framework to solve the multi-variate backfill problem, to build a novel real time anomaly detection classifier. We then demonstrate the use of this classifier in the states of Pennsylvania and California, with marked anomalies, with over 90% accuracy.

## 2 BACKGROUND

### 2.1 Problem Formulation

There are multiple accepted definitions of an anomaly when talking about COVID-19 reporting. Yom-Tov et al. [8] define an anomaly in spatiotemporal terms, referring to it as a rise in search terms in one region of the country that does not reflect in its neighboring regions. Jombart et al. [4] identify past temporal trends using linear regression, generalized linear models and Bayesian regression to detect outliers from the trend in a given season. Wang et al. [7] defined two different types of anomalies, one of which is based on outliers from a trend, given they are not a change point in the trend itself. The other definition refers to a case where the cumulative deaths at a given date are less than those at a previous date. Francisquini et al. [2] define an anomaly on a given day $t$ as an accentuated variation $r_i^t$ between the values reported at days $t-1$ and $t+1$ in a region $i$. This allows a differentiation between an anomaly, and a normal deviation from the previous trend and the beginning of a new one.

However, when attempting to detect anomalies in real-time, the above definitions do not help distinguish an anomaly from a change in trend since we do not have access to data from day $t+1$ at the time.

We consider anomalies in reporting, where initial reporting is incorrect due to human error, technical errors, and changes in standards and policies, but are later revised over time, called the 'backfill'

phenomenon. An example is the inclusion of probable deaths in the mortality reporting for the state of California, where data was back-distributed at a large scale over time before becoming stable. We define our task as supervised binary classification instead of the classical anomaly detection, to accommodate for the definitions of anomalous reports we use.

## 2.2 Back2Future

Back2Future[5] is a deep-learning model that attempts to solve the multi-variate backfill problem to refine real-time model predictions through Graph Recurrent Neural Networks by making use of a sequence of revisions to data reported for a particular day. It has experimentally been able to improve the real-time forecasts for all tested candidate models, highlighting how important it can be to incorporate backfill data. A backfill sequence (referred to as $B_{SEQ}$) is a series of revisions of a given signal made over time, ranging from the initial value as reported to the final or stable value.

This model attempts to solve two problems: the k-week ahead Backfill refinement problem, which estimates the stable value (the reported value for a timestep once no more revisions will be made to it) at a given target from a model, its historical predictions, and its current predictions for those targets. This means that given a model, and a history of its predictions, the goal is to improve the current prediction made by the model for k weeks ahead to better estimate the stable value of the target post any backfill adjustments[5]. The second problem it attempts to solve is the leaderboard refinement problem, which is a special case of the backfill refinement problem, attempting to predict the stable value of the current timestep based on current reports and historical reports, rather than k weeks ahead. This allows us to estimate the stable value of the ground-truth real-time data.

We introduce the usage of Back2Future to solve the leaderboard refinement problem and estimate the stable value for a given $B_{SEQ}$, and compare multiple $B_{SEQ}$s with the original values to build a supervised binary classification model to detect anomalies in real-time.

## 3 METHODOLOGY

We monitored the changelogs and the errata files, showing where the timeseries was refined retroactively, in the COVID-19 Data Repository by the Center for Systems Science and Engineering (CSSE) at Johns Hopkins University [1], aggregating them to the state level where necessary. We also collected the significance of changes in case counts reported in a single week, and the reason for this retrospective correction, if provided. As can be seen in Figures 1 and 2, there are some examples of such retrospective corrections as reported in the JHU dataset. We defined reporting anomalies in two different ways:

(1) Reports that have been corrected due to known factors like human error, change in reporting mechanisms or standards and backlogged cases introduced in a single day. We show an example of a sample anomaly in Pennsylvania, in the Chester county, where data was corrected on a large scale due to a misalignment between the county level and state level reporting.

(2) Reports of a state showing a retrospective correction of more than 3 standard deviations from the mean correction across that state over the experimental window from June 2020 to November 2021. Figure 2 shows an example from San Bernadino, California, where the number of cumulative deaths for February 25, 2021 was updated to 3,360 from 2,673, a change of more than 25% from the original reports on April 19, 2021 – more than a month after the original report.

June 2, 2021: Pennsylvania, US | Deaths for Chester county, PA were determined to be greater at the state level of reporting than the direct reporting from Chester county. The data has been redistributed based on the state level historical deaths and moving forward Chester county data will be collected from the maximum value between the state and county source

**Figure 1: Anomalous reporting found in data modification logs**

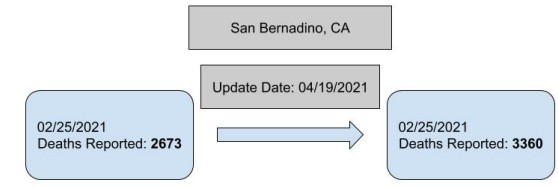

**Figure 2: Example of high correction in reporting in San Bernadino, CA**

Ground truth anomalies such as these cases were manually marked using these definitions using the logs and reports of modifications made to historical data. This included checking if there were major revisions to reports made in a region in a given week compared to the running mean of the reported values in the region, provided there was an explanation for these revisions based on the above definitions in the changelogs.

The historical data reports as of each week from 20th June 2020 to 27th November, 2021 were collected, and we ran the Back2Future neural framework on each time-series to solve the Leaderboard Refinement problem, as defined by Kamarthi et al. [5]. This gave us the reported values for each week in the time series, based on Back2Future's refinement from the initial reported value of each week. We then build a dataset consisting of the original reported values for mortalities and cases, the Back2Future (B2F) refined values, and whether the original report was considered an anomaly or not. Since multiple studies have shown that past records are often helpful to distinguish an anomaly from a change in disease trend (Wang et al. [7], Jombart et al. [4], Homayouni et al. [3]), we include the previous 4 weeks of reported and refined values in each

**Table 1: Features used in the Classifier**

| Feature | Description |
|---|---|
| *original_value* | Current week reported value |
| *b2f_corrected_value* | Current week refined value |
| *original_val_1prev* | 1 week previous reported value |
| *b2f_corrected_val_1prev* | 1 week previous refined value |
| *original_val_2prev* | 2 weeks previous reported value |
| *b2f_corrected_val_2prev* | 2 weeks previous refined value |
| *original_val_3prev* | 3 weeks previous reported value |
| *b2f_corrected_val_3prev* | 3 weeks previous refined value |
| *original_val_4prev* | 4 weeks previous reported value |
| *b2f_corrected_val_4prev* | 4 weeks previous refined value |

record of the dataset. This dataset is then used to train a Random Forest classifier to classify a record as anomalous or not as a binary classification task. Table 1 contains a list of all the features used to train the binary classification model. The target of this classifier is 'anomaly'. For this task, we chose to use a Random Forest classifier considering we were solving a supervised binary classification task, as opposed to the classical anomaly detection process.

## 4 EXPERIMENTS

We considered deaths in California and Pennsylvania for our experimental setup. We obtained data for 72 weeks, from 20th June 2020 to 27th November, 2021 and randomly split the records into training and testing data, maintaining a 70:30 split. The dataset contained 50 non-anomalous records and 22 anomalous records in Pennsylvania, and 62 non-anomalous records and 10 anomalous records in California. The random forest binary classifier trained resulted in an average accuracy of 90.90% in both states. Table 2 highlights some of the key metrics and Figure 3 shows the confusion matrix for the classifier in the state of Pennsylvania.

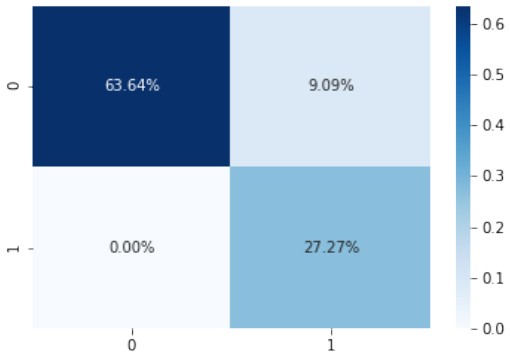

**Figure 3: Confusion Matrix for Classifier in PA**

This shows that this classifier when used with features generated by Back2Future, can result in accurate real time reporting anomaly detection.

**Table 2: Classifier Performance Metrics**

| Metric | PA | CA |
|---|---|---|
| **Accuracy** | 0.909 | 0.90 |
| **Precision** | 0.75 | 0.75 |
| **Recall** | 1.0 | 0.75 |
| **F1-Score** | 0.857 | 0.75 |

Furthermore, a study of the Gini importance of the features in trained model demonstrates that historical records are among the features that the model is most sensitive to. This study also shows that the Back2Future refined values for each historical record consistently have greater or equal importance to the corresponding original value for that week, validating the benefits of using the Back2Future framework for this experiment.

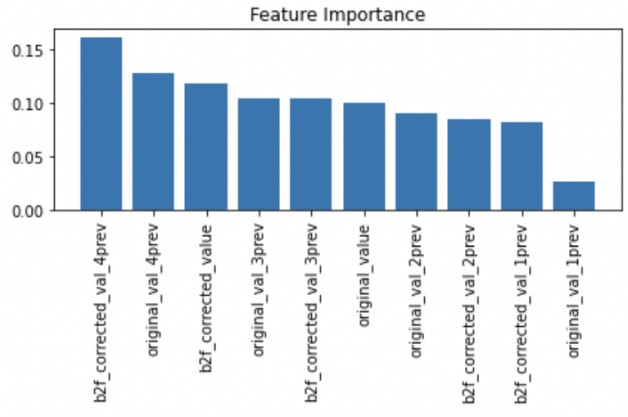

**Figure 4: Gini Importance for Features in Random Forest Classifier**

## 5 CONCLUSION

We have shown that we can make reliably detect anomalies in time-series for COVID-19 reporting in real-time. We demonstrate the use of a binary classifier on top of the Back2Future neural framework to detect reporting anomalies in US states, showing case studies of Pennsylvania and California. We believe that this can lead to further research in terms of incorporating other frameworks as features, thus making an even more robust anomaly detection tool. Further work can also be used for more granular anomaly detection at the county level as well as at a daily temporal scale. We also believe that studies exploring feature importance metrics like SHAP would further contribute to understanding underlying patterns in anomalies. While we approached this problem as a supervised binary classification task, there is potential for future study through formulating the problem as a multi-class classification problem to differentiate between types of anomalies as well. We can also utilize this mechanism in a forecasting model to methodologically demonstrate that removing anomalies can result in more reliable forecasting.

## ACKNOWLEDGMENTS

This paper is based on work partially supported by the NSF (Expeditions CCF-1918770, CAREER IIS-2028586, RAPID IIS-2027862, Medium IIS-1955883, Medium IIS-2106961, CCF-2115126), CDC MInD program, ORNL, faculty award from Facebook, and funds/ computing resources from Georgia Tech. We would also like to thank Harshavardhan Kamarthi and Alexander Rodríguez for their support and development of the Back2Future framework.

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
