# OpenReview forum: "Real-time Anomaly Detection in Epidemic Data Streams"
_ACM.org/SIGKDD/2022/Workshop/epiDAMIK — KDD 2022 Workshop epiDAMIK Poster_

### Official Review · Reviewer_H6K3 · 2022-06-25
**Interesting topic**

**Rating:** 3
**Confidence:** 4

**Review:**

This work describes an application of the Back2Future model to a COVID-19 mortality reporting dataset to identify anomalous mortality counts in real time. The experiments show that the model is able to classify new mortality reports as anomalous or not with high accuracy.

This paper has multiple strengths:
- Correcting reporting data in real time is an important area that could lead to important discussions in a data mining forum---good match for workshop.
- Solution generally seems reasonable: Backfill with Back2Future, then compare that model's predictions with observed values to get anomaly labels, then build a classifier to perform that prediction.

This paper could be improved with the following considerations:
- Slightly expanding the formal description of Back2Future's tasks (leaderboard refinement and backfilling) would be helpful to the reader and make the paper a little more self-contained.
- Expanding discussion around the dataset size would help. It's fine that the data are very small since it's at the weekly level, but it should be noted that anomalies are rare and how you account for this fact in the evaluation metrics. Further, at a glance, there seems to be a mismatch between using a complex model (Back2Future being a Graph Recurrent Neural Network) and a tiny dataset (72 timesteps?).
- Since the dataset is so small, it would be helpful to show that a really simple online anomaly detector doesn't work (motivating the need for a trained classifier). For example, one might monitor the different between new values and the running mean of a time series and set a threshold. If some of the labels were determined by finding big jumps in mortality reports, this anomaly detection algorithm is perfect for those instances!
- There is room to expand the discussion of the experiments. For instance, Figure 2 is just pointed to and Figure 3 isn't mentioned in the text.

Minor points:
- Line 95 has an extra "]"
- Line 131 has a floating "**here**"

---

### Official Review · Reviewer_Jus1 · 2022-06-25
**Interesting study but a very early report of a work in progress**

**Rating:** 2
**Confidence:** 4

**Review:**

The authors study the problem of detecting anomalies in real-time reports of COVID 19 data. They have built upon their previous work on retrospectively filling out past reports using a framework entitled Back2Future and used a Random Forest classifier to classify anomalous data.

There are several strong aspects of this work

- First, the problem being studied is of high importance. While the authors have mainly focused on COVID-19, past research [1] has identified such reporting issues for infectious diseases such as flu and the proposed solution can thus have a broad reaching impact
- The work builds on a strong `foundation` model which has been proven to work in correcting past reports. Using such corrected reports and the original reported numbers to identify anomalies is novel as well as has good motivations. One could envisage such a method being a natural part of a covid-19 forecasting pipeline
- The results (see more below) seems to show that the proposed solution has good performance for two states for the selected period of study

However, the paper in its current form may benefit from addressing the following aspects

- Experimental evaluation: The proposed approach seems to have been under analyzed. While the authors have used a training testing split, to accurately analyze a performance improvement, a cross validation based study would have been more useful. Especially, a time based cross validation [2] may be the appropriate study method. On a similar note, performance metrics are provided as point estimates without any confidence measures. This is especially important as the dataset is itself very small and without an uncertainty analysis its very hard to ascertain the importance of the performances.
- Lack of baselines, lack of motivation for classifier choice: Continuing on the previous thread, results have been shown without any baselines for comparisons. While one can argue for comparative baselines to select between the right kind of classifier (e.g. Random Forest vs GBT etc), even simple ad-hoc baselines could also be considered (e.g. rate of change of slop k sd above historical rate). Its also interesting to note that the authors have used RandomForest - a method more classically used for binary classification, and not specialized anomaly detection methods such as OneClassSVM - in general a motivation for the choice of classifier is lacking.
- Lack of Discussions and insights; Finally, the results haven't been interpreted in a satisfactory manner that could have led to gain insights. For  example,  the confusion matrix shows that for PA the method trades precision for a perfect recall while such trade-offs are not seen for CA - it would be interesting to analyze what drives these differences and perhaps study a few of the mis-classified instances.

Apart from the above, some of the other aspects that may need attention are as below:

- There are many formatting errors in the manuscript. Some instances are as below:

    * line 95: extract brackets around citation 8
    * line 128: citation is misplaced. Should be placed after the phrase that refers to the paper
    * line 131: misplaced `here`

- Variables have been introduced without proper definition and subsequent usage. For example, line 106, $r_i^t$ has been introduced without defining what $i$ refers to. Furthermore, this has not been used in subsequent part of the text. On the other hand, in line 127 $B_{SEQ}$ has been introduced as a variable which has been reused but without proper definitions.

- Consider using hyperlink while citing papers and referring to Figures/Tables e.g. line 154
- Figures 1 and 2 are largely uninformative and redundant given the description of the situations in the text. The space could have been used to report further experiments. Figure 3 doesn't have labels for the axes.
- line 161: unclear statement. consider avoiding the usage of `etc.`
- line 208: clarify how Back2Future was applied. Was it applied on the first reported value of the weeks or based of the values at a certain calendar date.
- line 211: `the reported values for each week in the time series, based` - should this reported or the refined values
- line  319: Use peer-cited versions of references. E.g. Reference 5 use the peer-reviewed published version instead of the arxiv version


**references in this review**

[1]: Chakraborty, Prithwish, et al. "What to know before forecasting the flu." PLoS computational biology 14.10 (2018): e1005964.
[2]: https://scikit-learn.org/stable/modules/cross_validation.html#time-series-split

---

### Official Review · Reviewer_smC3 · 2022-06-27
**More models, available dataset and state-of-the-art methodology**

**Rating:** 1
**Confidence:** 3

**Review:**

- The paper is not using any time series base model (RNN or other time series-based algorithms), but lag variables on random forest. This choice should be coming with an experiment (which could compare F1, and recall on different models).

- The paper does not try to highlight the benefits of using the b2f pre-trained model. What would be the output without these computed features?

- Moreover, as you are using a random forest classifier, you can easily compute the feature importance/sensibility, with algorithms such as Gini importance, Mean Decrease Accuracy, SHAP... This would be an important glimpse at what the model is using to make the classification.

- Only one year of data from two states is used. How does the model behave in California during the 2016 Wildfires?  Are you sure you are not overfitting?

- What is the behavior of the algorithm during outbreaks? You can use FEMA data to analyse your classifier (False positive)

---

### Official Review · Reviewer_ttBa · 2022-06-29
**Real-time Anomaly Detection in Epidemic Data Streams**

**Rating:** 4
**Confidence:** 3

**Review:**

This work addresses the problem of detecting anomalies in real-time in
time-series data, leveraging patterns found in data revisions that occur
retroactively. The authors apply Back2Future, a recently developed
GCN-based framework to address the Backfill refinement problem and the
Leaderboard refinement problem. In the Leaderboard refinement problem, we
are given forecasts from models made at timestep t-k, the current estimate,
and a revision dataset. The objective is to refine the current estimate and
in turn refine the evaluation of each model on the leaderboard. In this
work, the authors use this framework to first solve the leaderboard refinement
problem and estimate the stable value given revision dataset. Then, the
original time series and the refined time series are paired and labeled
with the information about whether an anomaly happened or not. These
feature vectors were used to train a random forest binary classifier.
Data from the COVID-19 Data repository by CSSE, JHU was used in this study.

Overall, this is a novel approach for real-time anomaly estimation given
that there are hardly any approaches in the literature, and very useful for
any forecast modeling. For training and validation purposes, anomalies in
the data have been detected by going through it in some detail. Patterns in
reporting anomalies have been identified and is a contribution in itself.

Random Forest classifier has built in parameter importance framework. The
authors could have used this facility to develop insight into the
sensitivity of the predictions to various parameters.